# TGF Beta as a Prognostic Biomarker of COVID-19 Severity in Patients with NAFLD—A Prospective Case–Control Study

**DOI:** 10.3390/microorganisms11061571

**Published:** 2023-06-13

**Authors:** Frano Susak, Nina Vrsaljko, Adriana Vince, Neven Papic

**Affiliations:** 1Department of Infectious Diseases, School of Medicine, University of Zagreb, 10000 Zagreb, Croatia; fsusak@student.mef.hr (F.S.); avince@bfm.hr (A.V.); 2Department for Viral Hepatitis, University Hospital for Infectious Diseases, 10000 Zagreb, Croatia; nvrsaljko@yahoo.com

**Keywords:** COVID-19, SARS-CoV-2, non-alcoholic fatty liver disease, NAFLD, TGF-β1, cytokines, inflammation

## Abstract

Non-alcoholic fatty liver disease (NAFLD), the leading cause of chronic liver disease in Western countries, has been identified as a possible risk factor for COVID-19 severity. However, the immunological mechanisms by which NAFLD exacerbates COVID-19 remain unknown. Transforming growth factor-beta 1 (TGF-β1) has an important immunomodulatory and pro-fibrotic role, which has already been described in NAFLD. However, the role of TGF-β1 in COVID-19 remains unclear, and could also be the pathophysiology link between these two conditions. The aim of this case–control study was to analyze the expression of TGF-β1 in COVID-19 patients depending on the presence of NAFLD and COVID-19 severity. Serum TGF-β1 concentrations were measured in 60 hospitalized COVID-19 patients (30 with NAFLD). NAFLD was associated with higher serum TGF-β1 concentrations that increased with disease severity. Admission TGF-β1 concentrations showed good discriminative accuracy in predicting the development of critical disease and COVID-19 complications (need for advanced respiratory support, ICU admission, time to recovery, development of nosocomial infections and mortality). In conclusion, TGF-β1 could be an efficient biomarker for predicting COVID-19 severity and adverse outcomes in patients with NAFLD.

## 1. Introduction

Since the outbreak of the COVID-19 pandemic, significant scientific efforts have been undertaken to identify risk factors and elucidate complex immune responses associated with COVID-19 clinical course and outcomes.

Non-alcoholic fatty liver disease (NAFLD) is the leading cause of chronic liver disease in developed countries, with a prevalence of about 25%, which parallels the prevalence of metabolic syndrome and its components [1]. The primary feature of NAFLD is steatosis, followed by nonalcoholic steatohepatitis (NASH) with inflammation, fibrosis and liver damage, eventually leading to cirrhosis and hepatocellular carcinoma (HCC). Its major pathological hallmarks include chronic low-level inflammation, impaired immune response, and endothelial dysfunction [2,3]. Importantly, NAFLD is now considered a multisystemic disease associated with a variety of extrahepatic complications, such as an increased cardiovascular risk, extrahepatic cancer or chronic kidney disease [4]. In the pre-COVID-19 era, the role of NAFLD in infectious diseases was largely overlooked [5,6]. However, in an attempt to better understand COVID-19 and find potential therapeutic options, the pandemic has yielded scientific research that has shed new light on the intertwined relationship between NAFLD and infection [7,8,9].

According to some studies, NAFLD is a significant risk factor for SARS-CoV-2 acquisition and hospitalization independently of other components of metabolic syndrome, possibly also linked with increased disease severity, prolonged hospital stays, and unfavorable outcomes [10,11,12]. However, the pathophysiological mechanisms by which NAFLD exacerbates COVID-19 remain unknown. One of the proposed hypotheses is that NAFLD aggravates the “cytokine storm” through the hepatic release of pro-inflammatory cytokines [13]. So far, research has shown that COVID-19 patients with NAFLD have a distinct cytokine profile including increased levels of IL-6, IL-8, IL-10, and CXCL10, all associated with a more severe clinical presentation [12,14,15,16,17]. 

Transforming growth factor-beta (TGF-β) is a pleiotropic cytokine involved in the regulation of cell proliferation, differentiation, migration, and apoptosis, playing a central role in preserving cellular homeostasis [18]. With its paradoxical dual role, promoting both immunosuppressive and pro-inflammatory effects, TGF-β is crucial for maintaining the immunological equilibrium [18,19]. Dysregulation of the TGF-β pathway is an important factor in the development and progression of liver disease [20]. Increased levels of TGF-β correlate with NAFLD severity and have been proposed as a marker of liver fibrosis [21,22]. Briefly, NAFLD begins with the accumulation of free fatty acids and triglycerides in the liver, leading to oxidative stress which induces the secretion of pro-inflammatory cytokines, including TGF-β [23]. Increased intrahepatic TGF-β levels are linked with hepatocyte death and activation and the conversion of hepatic stellate cells (HSCs) into myofibroblasts that secrete components of the extracellular matrix and induce fibrogenesis [20,23]. Excessive activation of the TGF-β signaling pathway over time causes hepatocyte resistance to its otherwise tumor-suppressive effect, which subsequently contributes to the development of HCC [20,22].

Research on TGF-β and COVID-19 is scarce, and the results are often contradictory. While some studies showed that TGF-β concentrations correlate with disease severity, others found no or even a negative correlation [24,25,26,27,28]. However, none of these studies examined the impact of TGF-β in patients with NAFLD, the patient group that constitutively has dysregulated TGF-β responses. 

Therefore, in this study, we hypothesize that patients with NAFLD have distinct TGF-β serum concentrations that might be used as a prognostic biomarker for COVID-19 severity.

## 2. Materials and Methods 

### 2.1. Study Design and Population

This monocentric study was conducted at the University Hospital for Infectious Diseases Zagreb (UHID), Croatia, and was part of the prospective cohort study that recruited patients with COVID-19 during the Delta wave of the pandemic (part of the COVID-FAT trial, ClinicalTrials.gov Identifier: NCT04982328, with the aim being to investigate the potential role and impact of NAFLD on COVID-19 severity and outcomes). 

Sixty adult SARS-CoV-2-positive patients consequently hospitalized during the period from April 2021 to December 2021 were included. SARS-CoV2 infection was confirmed in a nasopharyngeal swab sample using the real-time polymerase chain reaction (RT-PCR) method. 

The inclusion criteria were: COVID-19 requiring hospital admission and classified according to the National Institute of Health (NIH) guidelines in moderate, severe, or critical COVID-19 [29]. 

The exclusion criteria were: patients admitted to the ICU within 24 h of hospitalization or ones that died within the first 48 h of hospitalization, patients on corticosteroids before enrollment, and patients who had a bacterial infection at admission. Furthermore, patients with previously known chronic liver disease and cirrhosis, active malignancy, alcoholism, and pregnancy, patients that received parenteral nutritive support, immunocompromised patients including HIV-positive patients, and patients in palliative care were excluded.

Upon admission, patients were screened for components of metabolic syndrome, and abdominal ultrasound was performed by an experienced radiologist for the assessment of liver steatosis, as previously described [30]. The radiologist who performed ultrasound examination was blinded to the patient data. Patients were subsequently diagnosed with NAFLD according to current guidelines that require: (1) evidence of liver steatosis, (2) no excessive alcohol intake (as defined by an average consumption of >21 standard drinks per week in men or >14 standard drinks per week in women according to US standards [31]), (3) no competing causes of liver steatosis (including viral hepatitis, which was excluded by testing for HCV antibodies and HBsAg, and the use of medications associated with liver steatosis), and (4) no concurrent chronic liver disease [32]. Depending on the severity of the disease, according to the NIH classification, patients were classified into 3 subgroups—moderate (bilateral pneumonia with SpO_2_ > 93% on room air), severe (dyspnea and/or tachypnea > 24/min and/or SpO_2_ < 93%) and critical COVID-19 (requiring intensive care unit care, criteria for ARDS, advanced respiratory support with HFNC, non-invasive or invasive mechanical ventilation) [29]. 

Finally, 30 patients with NAFLD and 30 without NAFLD were included in the study. The sample size of 30 patients was selected according to power analysis for the Mann–Whitney U test to achieve an 80% chance of detecting a difference in median TGF-β1 concentrations between the two groups (NAFLD vs. non-NAFLD) at a 5% significance level. 

The included participants had not been part of previous studies, and all provided written informed consent. The study was approved by the School of Medicine, the University of Zagreb Ethics Committee, and the Ethics Committee of the UHID Zagreb (code 01-673-4-2021).

### 2.2. Laboratory and Clinical Data

At the time of hospital admission, baseline patient characteristics, including demographic data, comorbidities, chronic medication use, baseline clinical status, and vital parameters were collected. 

As part of the standard diagnostic procedure, routine laboratory tests were collected, including: hemoglobin, red blood cell count (RBC), platelet count (Plt), white blood cell count (WBC), absolute neutrophil and lymphocyte count (ANC and ALC, respectively), blood urea nitrogen (BUN), serum creatinine, bilirubin, aspartate aminotransferase (AST), alanine aminotransferase (ALT), gamma-glutamyl transferase (GGT), lactate dehydrogenase (LDH), alkaline phosphatase (ALP), prothrombin time (PT), fibrinogen, C-reactive protein (CRP), procalcitonin (PCT), ferritin and interleukin 6 (IL-6). 

Anthropometric measurements, including body mass index (BMI), waist circumference (WC), and waist–hip ratio (WHR), were documented.

Ultrasound was used for measuring the visceral and subcutaneous abdominal fat thickness. Subcutaneous fat was measured at three sites on the abdominal wall: (1) between the xiphoid and umbilicus, (2) the right mid-axillary line superior to the iliac crest, and (3) the left mid-axillary line superior to the iliac crest. Visceral fat was approximated by measuring the perirenal adipose tissue, as previously described [33]. 

### 2.3. TGF-β1 Measurments

At hospital admission, an additional blood sample was taken from the patients, and serum concentration of transforming growth factor beta 1 was determined using the TGF-β1 Human ProcartaPlex Simplex Kit (Thermo Fisher Scientific, Waltham, MA, USA). Two measurements were obtained, and the mean value was used in the analysis.

### 2.4. Statistical Analysis

The clinical characteristics, laboratory, and demographic data were evaluated and descriptively presented as frequencies and medians with interquartile ranges. Fisher’s exact test and the Mann–Whitney U test were used to compare the two groups. The Kruskal–Wallis test with Dunn’s multiple comparisons test was used to compare three or more groups. All tests were two-tailed; a *p*-value < 0.05 was considered statistically significant. Correlations were analyzed using Spearman’s rank correlation coefficient and summarized in a correlation matrix. The discriminatory performances of the laboratory variables considered were compared using receiver operating characteristic (ROC) analysis. Time to hospital discharge or readiness for discharge stratified by biomarker levels was evaluated using the Kaplan–Meier method and hazard ratios (HR) with 95% confidence intervals (95%CI) and *p*-values were calculated via the log-rank test. Risk factors associated with mortality were investigated using a univariate and subsequently multivariable logistic regression analysis. The strength of association was expressed as an odds ratio (OR) and its corresponding 95%CI. Statistical analyses were performed using GraphPad Prism Software version 9.5.1 (San Diego, CA, USA).

## 3. Results

### 3.1. Baseline Patients’ Characteristics

Of the 60 included patients, 30 patients were classified into the NAFLD group (53.33% males; the median age of 56, IQR 36–65) and 30 were classified into the non-NAFLD group (66.67% males; the median age of 64, IQR 36–70). There were no differences in demographic and comorbidity data, chronic medications and baseline clinical status between the two groups, as presented in Table 1. The median time interval from disease onset to hospital admission was similar between the groups (9, IQR 7–11 vs. 9.5, IQR 7.3–11, *p* = 0.717). There was, however, a statistically significant difference in anthropometric measurements; the NAFLD group had a higher BMI (31 kg/m^2^, IQR 26–34 vs. 28 kg/m^2^, IQR 24–31, *p* = 0.015) and a higher WHR (1.0, IQR 0.96–1.1 vs. 0.98, IQR 0.83–1.1, *p* = 0.030). Additionally, the NAFLD group had higher surrogate markers of visceral adiposity-perirenal fat (12 mm, IQR 8–14 vs. 6.5 mm, IQR 5–8, *p* < 0.0001) measured via ultrasound, and thicker subcutaneous fat (24 mm, IQR 17–28 vs. 18 mm, IQR 13–21, *p* = 0.005). 

Laboratory findings at admission are shown in Table 2. Patients in the NAFLD group had a higher leukocyte count (7.4 × 10^9^/L, IQR 5.6–10 vs. 5.7 × 10^9^/L, IQR 4.8–7.1, *p* = 0.028), platelet count (219 × 10^9^/L, IQR 160–270 vs. 176 × 10^9^, IQR 121–236, *p* = 0.011), gamma-glutamyl transferase level (61 IU/L, IQR 42–131 vs. 38 IU/L, IQR 24–62, *p* = 0.018), C-reactive protein level (140 mg/L, IQR 71–221 vs. 90 mg/L, IQR 39–139, *p* = 0.032), and IL-6 level (87 pg/L, IQR 15–150 vs. 41 pg/L, IQR 14–62, *p* = 0.035).

There was no difference in the median duration of hospitalization between the two groups (9.5, IQR 7–17 vs. 9.0, IQR 5.8–13 days, *p* = 0.5147). Overall, 15 patients were admitted to the ICU (8 (26.67%) in the NAFLD group and 7 (23.33%) in the non-NAFLD group) and there were no differences in mortality (7 patients in NAFLD and 3 in the non-NAFLD group died, *p* = 0.299).

### 3.2. Association of Serum Concentrations of TGF-β1 with COVID-19 Severity in Patients with and without NAFLD

At admission, TGF-β1 was significantly higher in patients with NAFLD (13,221 pg/mL, IQR 12,148–16,128 vs. 9447 pg/mL, IQR 6518–10,758, median difference 5599, 95%CI 2731 to 8468 pg/mL, *p* < 0.0001). 

Next, we assessed the association of serum TGF-β1 concentrations upon admission with subsequent disease progression during hospitalization (Figure 1). Patients were divided according to the previously mentioned NIH classification of COVID-19 severity [29]; in the NAFLD group, 9 patients developed critical, 14 developed severe, and 7 developed moderate COVID-19. Of 30 patients without NAFLD, 7 developed critical, 14 developed severe, and 9 developed moderate COVID-19.

Within the NAFLD group, those with critical COVID-19 had significantly higher TGF-β1 concentrations upon admission compared to those with moderate COVID-19 (14,848 pg/mL, IQR 13,352–21,491 vs. 10,601 pg/mL, IQR 10,112–13,199, *p* = 0.0458). In the non-NAFLD group, there were no significant differences in measured TGF-β1 concentrations depending on COVID-19 severity, although there was a trend towards higher concentrations in more severe forms of the disease. Both patients with severe and critical COVID-19 and NAFLD had higher TGF-β1 serum concentrations than patients without NAFLD.

Next, using receiver operating characteristic (ROC) curve analysis, we calculated the area under the curve (AUC) of TGF-β1 as a diagnostic biomarker for distinguishing subgroups of patients depending on COVID-19 severity, as shown in Figure 2. Serum concentrations of TGF-β1 higher than 10,400 pg/mL had 66.67% sensitivity (95%CI 54.46% to 76.99%) and 66.67% specificity (95%CI 48.20% to 83.87%) when differentiating moderate from critical and severe COVID-19 with an AUC of 0.72 (95%CI 0.61–0.84, *p* = 0.0092). Similarly, TGF-β1 concentrations higher than 11,400 pg/mL had 68.75% sensitivity (95%CI 48.20% to 83.87%) and 61.36% specificity (95%CI 48.97% to 72.44%) when differentiating critical from less severe forms of COVID-19 with AUC of 0.7116 (95%CI 0.54–0.83, *p* = 0.0127).

### 3.3. Correlation Analysis of Serum TGF-β1 Concentrations in Patients with COVID-19

Next, we analyzed potential correlations between serum TGF-β1 concentrations and other clinical and laboratory parameters in patients with COVID-19, as presented in Figure 3. Serum TGF-β1 concentrations correlated positively with several inflammatory biomarkers; CRP (r = 0.35, *p* = 0.008), leukocyte count (r = 0.36, *p* = 0.005), absolute neutrophile count (r = 0.28, *p* = 0.003), platelet count (r = 0.46, *p* < 0.0001), platelet/lymphocyte ratio (r = 0.25, *p* = 0.05), and fibrinogen (r = 0.33, *p* = 0.01). A significant negative correlation was shown with creatinine (r = −0.24, *p* = 0.034). TGF-β1 measurements also demonstrated a positive correlation with the thickness of subcutaneous (r = 0.38, *p* = 0.0003) and visceral (r = 0.28, *p* = 0.032) fat, while there were no correlations with BMI or WHR. Other laboratory and clinical parameters, such as age or duration of hospitalization, showed no correlation with serum concentrations of TGF-β1.

Furthermore, as presented in Figure 4, there were no differences in serum TGF-β1 concentrations in subgroups of patients according to T2DM, obesity, dyslipidemia, duration of symptoms before admission and age group. 

### 3.4. Association of Serum TGF-β1 Concentrations with COVID-19 Clinical Outcomes and Complications

As previously mentioned, there were no differences in hospitalization duration, need for oxygen support, ICU admission, or the development of pulmonary thrombosis between the NAFLD and non-NAFLD groups.

We examined the impact of serum TGF-β1 concentrations on time to recovery, as defined by time to hospital discharge or readiness for discharge. In survival analysis using Kaplan–Meier estimates, a TGF-β1 concentration higher than 12,000 pg/mL (HR 2.05, 95%CI 1.07 to 3.96, log-rank test *p* = 0.03) appeared to be an efficient prognostic biomarker associated with longer time to recovery, as presented in Figure 5.

We further assessed the association of measured TGF-β1 upon admission with clinical outcomes and complications of COVID-19 during hospitalization, as shown in Figure 6. 

In our cohort, 19 patients required advanced oxygen therapy (which includes HFNC, NIV, and IMV). These patients at admission had higher serum concentrations of TGF-β1 compared to those who required no or only low-flow supplemental oxygen (12,033 pg/mL, IQR 10,507–18,014 vs. 10,216 pg/mL IQR 7521–13,384, *p* = 0.0209, mean difference 2681, 95%CI 580 to 5591).

Regarding the admission to the ICU, 15 patients that required ICU treatment had significantly higher serum concentrations of TGF-β1 (11,774 pg/mL, IQR 11,123–18,270 vs. 10,543 pg/mL, IQR 8253–13,384, *p* = 0.0379, mean difference 2656, 95%CI 368 to 6255)

Of 60 hospitalized patients, 12 developed PT; 7 in the NAFLD group and 5 in the non-NAFLD group. There were no significant differences in the concentrations of serum TGF-β1 between those who developed PT and those without PT (11,907 pg/mL, IQR 9202–18,306 vs. 10,112 pg/mL, IQR 7226–13,684, *p* = 0.2373). 

Next, we analyzed the association of measured levels of TGF-β1 with the risk of developing bacterial superinfection as a complication of COVID-19. A total of 11 patients developed bacterial complications during hospitalization: 9 with NAFLD and 2 without NAFLD. Serum concentrations of TGF-β1 were significantly higher in patients with bacterial superinfection (14,459 pg/mL, IQR 11,123–20,794 vs. 10,601 pg/mL, IQR 8626–13,221, *p* = 0.0115, mean difference 4827, 95%CI 1125 to 8575).

### 3.5. Association of Serum TGF-β1 Concentrations with COVID-19 Mortality

Finally, we examined the impact of measured TGF-β1 concentrations upon admission on COVID-19 mortality. During the hospital stay, 10 patients died, 7 of whom had NAFLD. As shown in Table 3, patients who died had higher serum concentrations of TGF-β1 (13,352 pg/mL, IQR 11,090–19,250 vs. 10,572 pg/mL, IQR 8731–20,794, *p* = 0.0192). There were no significant differences in other laboratory biomarkers, such as ferritin, CRP, and LDH, except for IL-6, which was also higher in patients who died (95 pg/L, IQR 49–164 vs. 41 pg/L, IQR 6.9–90, *p* = 0.0262). 

Using ROC curve analysis, we calculated the AUC of TGF-β1 as a diagnostic biomarker for COVID-19 mortality, as shown in Figure 7. Concentrations of TGF-β1 higher than 12,000 pg/mL showed 60% sensitivity (95%CI 35.16% to 80.58%) and 63% specificity (95%CI 51.52% to 73.62%) for COVID-19 mortality with an AUC of 0.73 (95%CI 0.61 to 0.86, *p* = 0.0153). Similarly, IL-6 had an AUC of 0.72 (95%CI 0.59 to 0.86, *p* = 0.0273), with serum concentrations higher than 60 pg/mL having 70% sensitivity (95%CI 44.17% to 87.31%) and 60% specificity (95%CI 47.02% to 71.71%) for COVID-19 mortality.

In order to identify factors associated with mortality, we performed multivariable logistic regression analysis, which identified baseline TGF-β1 > 12,000 pg/mL (OR 2.06, 95%CI 1.51–13.2, *p* = 0.006) and the need for ICU admission (OR 6.08, 95%CI 1.55–29.32, *p* = 0.145) associated with mortality (AUC 0.83, 95%CO 0.69–0.97). Age, sex, the presence of obesity, T2DM, CRP, LDH and IL-6 were not associated with mortality in our model. 

## 4. Discussion

Here, we provide the first evidence that patients with COVID-19 and NAFLD have higher TGF-β1 serum concentrations that continue to increase with disease severity. On the contrary, in the non-NAFLD group, there was no significant correlation between TGF-β1 concentrations and disease severity. These findings were not associated with other components of metabolic syndrome, including T2DM, obesity, or hyperlipidemia. Furthermore, serum TGF-β1 concentrations appear to be an efficient prognostic biomarker associated with time to recovery, need for advanced respiratory support, ICU admission and the development of nosocomial infections in COVID-19 patients. In the current study, although there was trend towards higher mortality in patients with NAFLD, it was not statistically significant. However, this study was not designed to analyze the impact of NAFLD on mortality, but rather differences in TGF-β1 serum profiles.

So far, several studies have reported a distinct immune response in patients with NAFLD during SARS-CoV-2 infection. We recently showed that patients with NAFLD have higher levels of IL-6, IL-8, IL-10, and CXCL10, and lower levels of IFN-γ, and these were associated with worse COVID-19 outcomes [16]. However, most of the studies analyzed the same cytokine panel and none of them examined TGF-β1 in this patient population. 

TGF-β1 is a pleiotropic cytokine involved in various processes such as cellular proliferation, immune tolerance, wound healing, fibrosis, and tumor suppression [34]. With its primarily anti-inflammatory role, TGF-β1 inhibits effector immune cells such as T cells, B cells, NK cells, and dendritic cells [18,35]. On the other hand, TGF-β1 can also drive inflammation by promoting the differentiation of Th17 and Th22 cells [19]. The controlled activity of TGF-β1 is essential for the immunological equilibrium; while TGF-β1 enables healing by silencing immune responses and preventing uncontrolled inflammation, TGF-β1 acts as one of the main mediators and initiators of fibrosis with apoptotic effects on endothelial cells that, if not controlled, could cause tissue damage [34,35]. Importantly, TGF-β1’s role in infections is highly tissue- and pathogen-dependent. 

The role of TGF-β1 has already been investigated in other viral respiratory infections. A study conducted by Rendón-Ramirez et al. showed that patients with influenza A (H1N1) had significantly lower concentrations of TGF-β1 compared to healthy controls [36]. In vitro experiments showed that in the context of influenza A infection, epithelial-derived TGF-β1 has a strong pro-viral effect by suppressing early immune response during infection [37]. Likewise, it was shown in vivo that neuraminidases of the influenza A virus through TGF-β1 activation enhance the expression of cellular adhesins, leading to increased bacterial loading in the lungs [38], which partly explains the frequent bacterial complications during influenza infection. 

In contrast, in SARS, TGF-β1 is elevated in plasma and lung tissues in patients during the early phase of SARS-CoV infection, which is associated with enhanced viral replication, and the development and progression of lung fibrosis [39,40]. 

Surprisingly, the role of TGF-β1 in COVID-19 is less clear, and the published results are often conflicting. In a prospective case–control study conducted by Ghazavi et al., higher serum concentrations of TGF-β1 among COVID-19 patients were reported [41]. Similarly, higher TGF-β1 concentrations were reported in a Turkish COVID-19 cohort in a study that compared 50 COVID-19 patients to 45 healthy controls [42]. In both studies, the levels of TGF-β1 correlated with disease severity. On the other hand, Karadeniz et al., in their cross-sectional study, showed no significant difference in TGF-β1 concentrations between 59 COVID-19 patients and 30 healthy controls [27]. Recently, Zivancevic-Simonovic S et al. showed that lower serum TGF-β1 concentrations were associated with unfavorable outcomes in severe COVID-19 [43]. In patients with long COVID who developed pulmonary fibrosis, TGF-β1 was elevated [44], which is not unusual considering that TGF-β is one of the main promoters of fibrosis [24]. 

These discrepancies among studies could not be fully explained by variations among different assays and the lack of standardization for measuring cytokines, but patient selection, specific comorbidities that might influence TGF-β responses, the timing of cytokine measurement or the circulating SARS-CoV2 variant could have an important role as well. Indeed, in the non-NAFLD group, we found no significant correlation between TGF-β1 and COVID-19 severity, while this association was significant in NAFLD patients. 

While the TGF-β signaling pathway has been associated with the development of components of metabolic syndrome [45], the levels of TGF-β1 have not been sufficiently explored during COVID-19 in these patient subgroups. A study by El-Din et al. included 70 COVID-19 patients and showed that the obese diabetic group had significantly higher TGF-β1 concentrations in comparison to the non-obese non-diabetic group [46]. However, they found no difference in TGF-β1 concentrations in obese diabetic patients who developed ARDS compared to patients who did not [46]. Similarly, single-cell transcriptome analysis reported blunted cytokine responses in macrophages and dendritic cells isolated from obese COVID-19 patients, except for TGF-β1, which was broadly expressed in the obese group [47]. 

There are several possible roles of TGF-β1 in COVID-19 pathogenesis. TGF-β1 suppresses innate and adaptive immune response by inhibiting NK cells [48] and causing ineffective IgA class switching [26], resulting in uncontrolled viral replication and more severe forms of the disease. Most unfavorable outcomes of COVID-19 are accompanied by ARDS. TGF-β1 plays a role in both the early and terminal stages of ARDS; it inhibits alveolar fluid reabsorption via endocytosis of the apical sodium channels in type 2 pneumocytes [49], and as a main pro-fibrotic factor, TGF-β1 drives tissue repair and lung remodeling [24]. TGF-β1-induced altered immune response and subsequent disease progression are in line with our findings where serum TGF-β1 concentrations correlated with disease severity, time to recovery, need for advanced respiratory support, ICU admission and mortality.

There are several possible explanations for why patients with NAFLD have elevated TGF-β1 concentrations. TGF-β1 plays an important role in all stages of NAFLD; most prominently, it causes hepatocytes apoptosis and differentiation of hepatic stellate cells (HSC) to extracellular matrix-secreting myofibroblasts [22,23,50,51]. TGF-β1 is natively elevated in patients with NAFLD, and its serum concentrations also correlate with progression to NASH and have been proposed as a marker of liver fibrosis [21,22]. 

Our study should be viewed within its limitations. We included a relatively small sample size of 60 patients, the diagnosis of NAFLD was based on abdominal ultrasound, and the TGF-β1 concentrations were examined only at admission. Due to the exclusion criteria, patients who died within 48 h of hospital admission were not included and the TGF-β1 levels in the most severely ill patients were not evaluated. Since scores for fibrosis may not be reliable in hospitalized patients with COVID-19, which is frequently accompanied by elevated aminotransferases, and elastography was not available at the COVID-19 department, we could not evaluate the fibrosis stage; consequently, the effect of advanced NAFLD on TGF-β1 levels could not be analyzed. Since this was an observational study, causality could not be determined; results cannot be generalized due to the limited study population. The small number of participants in COVID-19 severity subgroups also limits the statistical significance of the obtained results.

Nevertheless, we studied a well-defined cohort of patients and report the first data examining the TGF-β1 profile in patients with NAFLD and COVID-19. Longitudinal studies are needed to further evaluate TGF-β1 levels and their significance in patients with components of metabolic syndrome and NAFLD. 

## 5. Conclusions

In conclusion, we have shown that patients with NAFLD have increased serum TGF-β1 concentrations that correlate with disease severity and outcomes. This could partly explain the higher risk of COVID-19 complications in this patient population. Considering the overlapping of both NAFLD and the COVID-19 pandemic, additional research is needed to cultivate a better understanding of the pathophysiology and underlying immunological mechanisms in this group of patients, leading to new prognostic biomarkers and a more individualized therapeutic approach. 

## Figures and Tables

**Figure 1 microorganisms-11-01571-f001:**
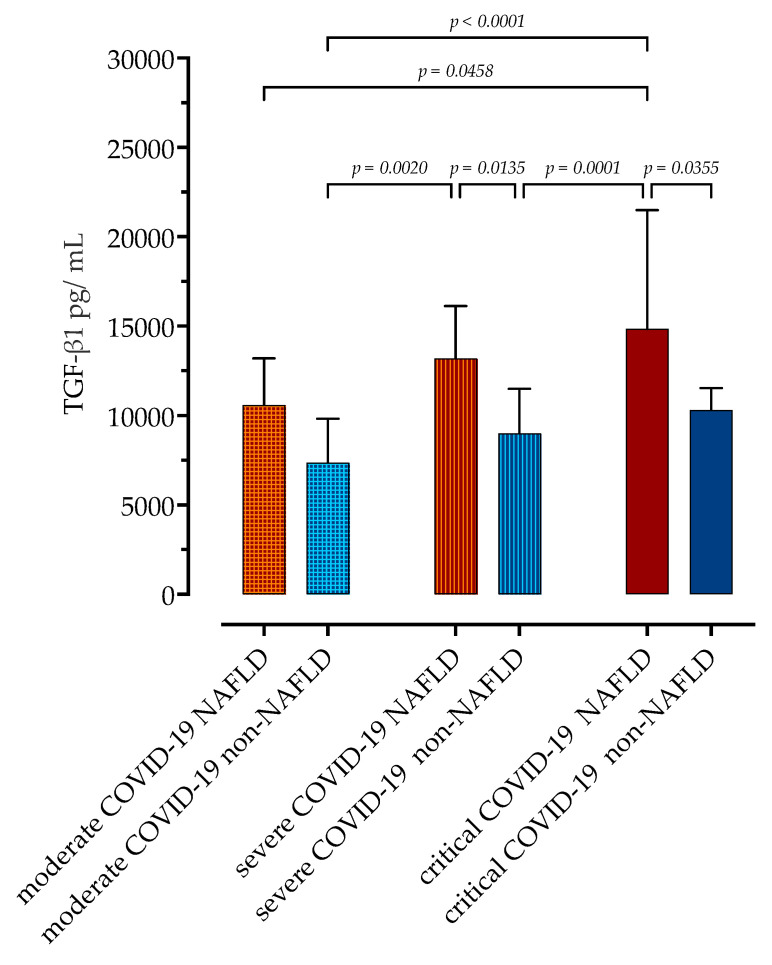
Serum concentrations of TGF-β1 in patients with and without NAFLD depending on the COVID-19 severity. Data are presented as medians with IQRs and analyzed via a Kruskal–Wallis test with Dunn’s multiple comparisons test.

**Figure 2 microorganisms-11-01571-f002:**
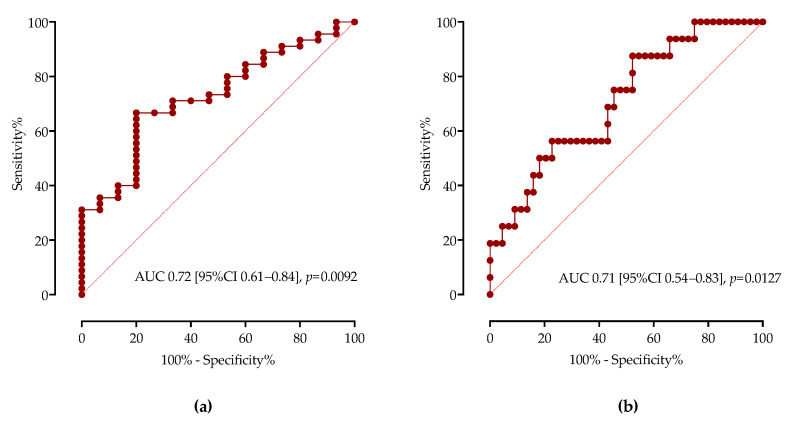
The ROC curve analysis of serum TGF-β1 concentrations for discrimination of: (**a**) moderate COVID-19 and severe/critical COVID-19 patients; (**b**) critical COVID-19 and moderate/severe COVID-19. AUCs are shown with 95%CI.

**Figure 3 microorganisms-11-01571-f003:**
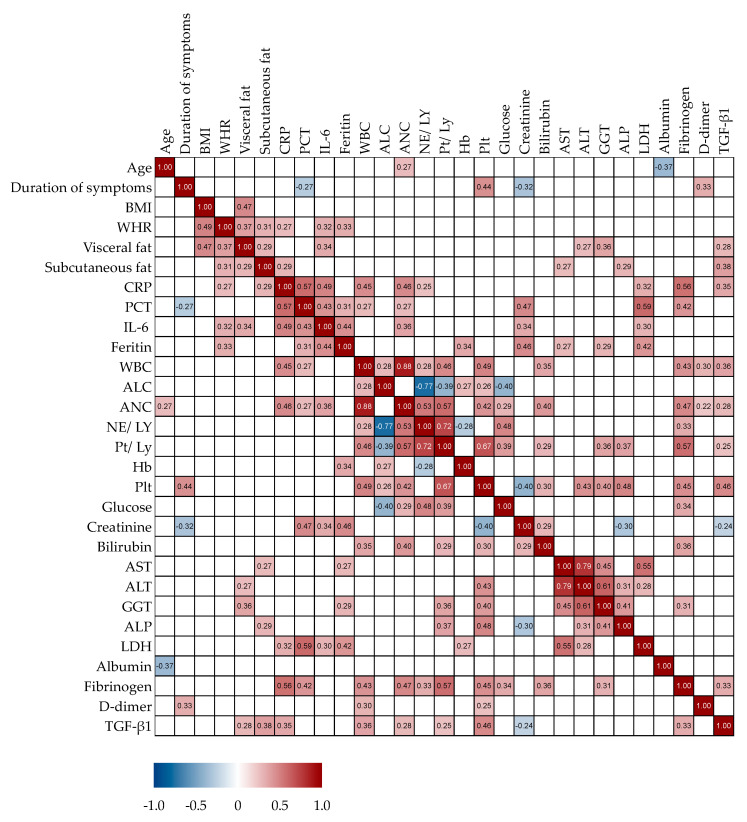
Spearman’s correlation correlogram. The strength of the correlation between two variables is represented by the color at the intersection of those variables. Colors range from dark blue (strong negative correlation; r = −1.0) to red (strong positive correlation; r = 1.0). Results were not represented if *p* > 0.05.

**Figure 4 microorganisms-11-01571-f004:**
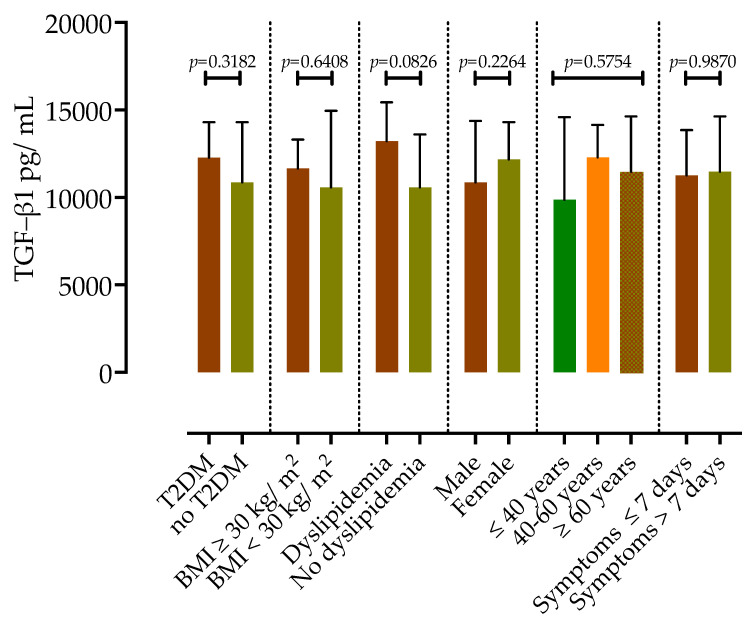
TGF-β1 serum concentrations in COVID-19 patients according to the presence of type 2 diabetes mellitus (T2DM), obesity, dyslipidemia, sex, age group and duration of symptoms. Data are presented as medians with IQR, and *p*-values were calculated via the Mann–Whitney test or Kruskal–Wallis test.

**Figure 5 microorganisms-11-01571-f005:**
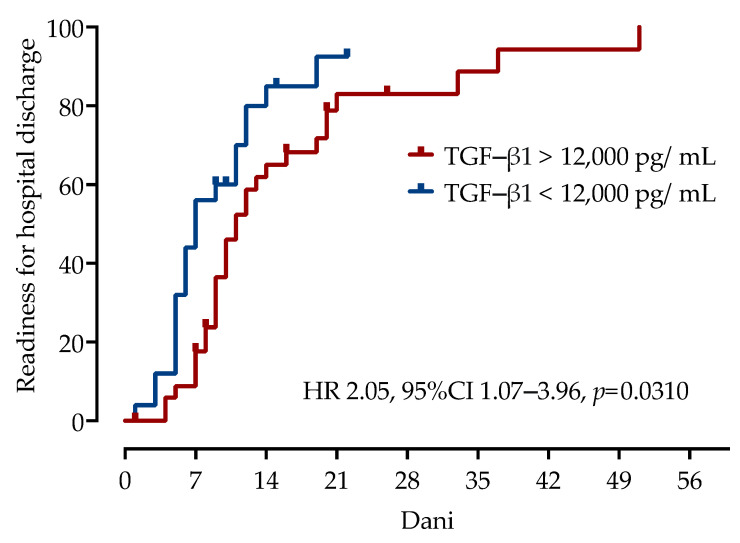
Association of time to recovery with serum concentrations of TGF-β1 using Kaplan–Meier curves in patients with COVID-19. Hazard ratios with 95% confidence intervals and *p*-values were calculated using the log-rank test.

**Figure 6 microorganisms-11-01571-f006:**
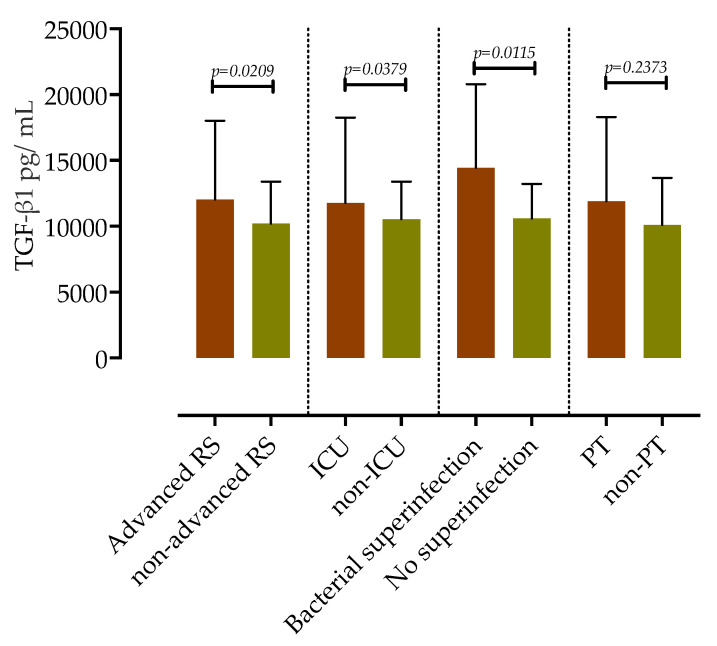
TGF-β1 serum concentrations in COVID-19 patients according to the level of respiratory support, ICU admission, development of nosocomial infections, and pulmonary thrombosis. Data are presented as medians with IQR, and *p*-values were calculated using the Mann–Whitney test.

**Figure 7 microorganisms-11-01571-f007:**
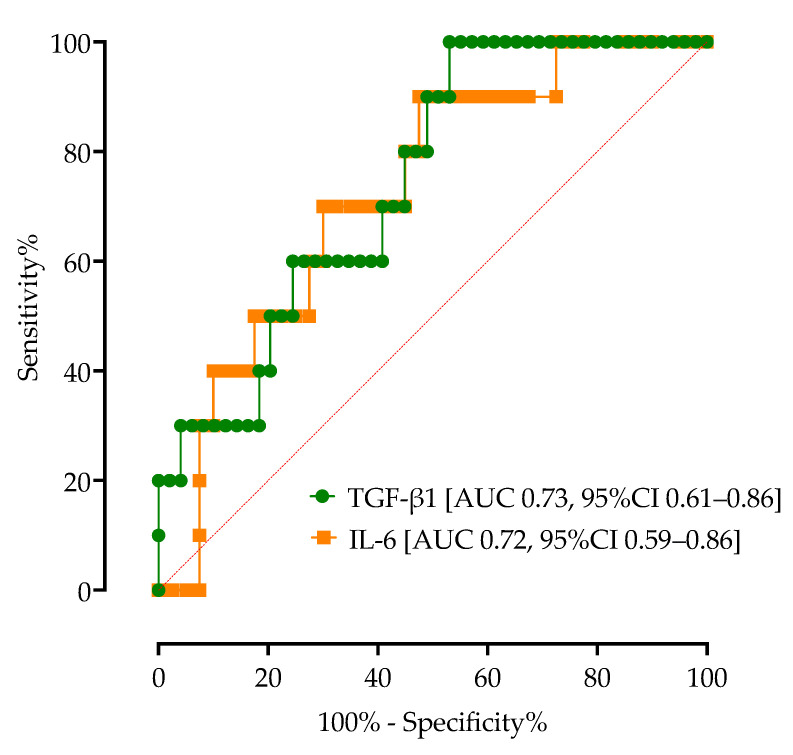
ROC curve analysis of TGF-β1 and IL-6 with the corresponding AUCs for COVID-19 mortality.

**Table 1 microorganisms-11-01571-t001:** Baseline patients’ characteristics.

	NAFLD (*n* = 30)	Non-NAFLD (*n* = 30)	*p*-Value ^a^
Age, median (IQR)	56 (36–65)	64 (36–70)	0.297
Male, No. (%)	16 (53.33%)	20 (66.67%)	0.429
Comorbidities			
Diabetes Mellitus	6 (20.00%)	6 (20.00%)	1.000
Arterial Hypertension	16 (53.33%)	17 (56.67%)	1.000
Obesity	16 (53.33%)	10 (33.33%)	0.192
Gastritis/GERD	3 (10.00%)	1 (3.33%)	0.612
Cardiovascular disease	3 (10.00%)	2 (6.67%)	1.000
Clinical findings on admission			
Peripheral oxygen saturation (SpO_2_), %	93 (88–96)	93 (90–95)	0.828
Respiratory rate,/min	24 (22–31)	24 (18–28)	0.141
Heart rate,/min	93 (83–103)	95 (81–100)	0.828
Body temperature, °C	38 (37–39)	38 (37–38)	0.076
Anthropometric and ultrasound measurements			
BMI, kg/m^2^	31 (26–34)	28 (24–31)	0.015
Waist-hip ratio (WHR)	1.0 (0.96–1.1)	0.98 (0.83–1.1)	0.030
Subcutaneous fat, mm	24 (17–28)	18 (13–21)	0.005
Visceral fat, mm	12 (8–14)	6.5 (5–8)	<0.0001

^a^—Fisher exact or Mann–Whitney U test, as appropriate. Data are presented as medians (interquartile range) or frequencies (percentage). Abbreviations: IQR—interquartile range, GERD—gastroesophageal reflux disease, SpO_2_—peripheral oxygen saturation, BMI—body mass index, WHR—waist–hip ratio.

**Table 2 microorganisms-11-01571-t002:** Laboratory findings on admission.

	NAFLD (*n* = 30)	Non-NAFLD (*n* = 30)	*p*-Value ^a^
Leukocyte count, 10^9^/L, median (IQR) ^b^	7.4 (5.6–10)	5.7 (4.8–7.1)	0.028
Lymphocyte count, 10^9^/L	0.73 (0.51–1.1)	0.63 (0.44–0.81)	0.248
Neutrophil count, 10^9^/L	5.9 (4.1–8.8)	4.6 (4.1–5.9)	0.097
Neutrophils/lymphocytes ratio	8 (4–14)	7 (5.3–10)	0.889
Hemoglobin, g/L	138 (129–146)	134 (124–148)	0.447
Platelet count, 10^9^/L	219 (160–270)	176 (121–236)	0.011
Platelets/lymphocytes ratio (PTR)	16 (13–37)	15 (10–22)	0.197
Urea, mmol/L	5.7 (4.4–7.2)	5.7 (4.3–9.5)	0.689
Creatinine, μmol/L	72 (63–94)	79 (63–99)	0.511
AST, IU/L	47 (35–79)	46 (35–85)	0.738
ALT, IU/L	39 (27–86)	38 (21–58)	0.154
GGT, IU/L	61 (42–131)	38 (24–62)	0.018
ALP, IU/L	59 (52–86)	55 (44–76)	0.293
LDH, IU/L	401 (275–560)	415 (283–510)	0.933
Bilirubin, μmol/L	12 (8.8–14)	11 (9–14)	0.997
Prothrombin time	1.2 (1.1–1.3)	1.2 (1–1.3)	0.892
Fibrinogen, g/L	6.0 (5.4–6.9)	5.7 (5.1–6.6)	0.425
D-dimers, mg/L	1.1 (0.54–2)	0.93 (0.58–1.2)	0.425
CRP, mg/L	140 (71–221)	90 (39–139)	0.032
Procalcitonin μg/L	0.18 (0.077–0.41)	0.14 (0.069–0.32)	0.539
IL-6, pg/L	87 (15–150)	41 (14–62)	0.035
Ferritin, μg/L	1111 (706–1797)	711 (505–1564)	0.207
APRI score	0.72 (0.51–1.3)	0.60 (0.34–0.91)	0.119
FIB-4 score	3.3 (1.7–4.4)	2.0 (1.2–2.7)	0.009

^a^—Fisher exact or Mann–Whitney U test, as appropriate; ^b^—data are presented as medians (interquartile range). Abbreviations: IQR—interquartile range, AST—aspartate aminotransferase, ALT—alanine aminotransferase, GGT—gamma-glutamyl transferase, ALP—alkaline phosphatase, LDH—lactate dehydrogenase, CRP—C-reactive protein, IL-6—interleukin-6, APRI—AST to Platelet Ratio Index.

**Table 3 microorganisms-11-01571-t003:** Serum concentrations of laboratory biomarkers and TGF-β1 in patients with COVID-19 depending on their outcome.

	Deceased (*n* = 10)	Survived (*n* = 50)	*p*-Value ^a^
TGF-β1, pg/mL, median (IQR) ^b^	13,352 (11,090–19,250)	10,572 (8731–20,794)	0.0192
IL-6, pg/L	95 (49–164)	41 (6.9–90)	0.0262
Ferritin, μg/L	948 (679–1797)	743 (553–1591)	0.6015
CRP, mg/L	129 (80–162)	103 (56–188)	0.7324
LDH, IU/L	436 (330–614)	399 (273–512)	0.3604

^a^—Mann–Whitney U test; ^b^—data are presented as medians (interquartile range). Abbreviations: IQR—interquartile range, TGF-β1—transforming growth factor-beta1, IL-6—interleukine-6, CRP—C-reactive protein, LDH—lactate dehydrogenase

## Data Availability

The datasets generated during and/or analyzed during the current study are available from the corresponding author on reasonable request.

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
