# Peer review of "TGF Beta as a Prognostic Biomarker of COVID-19 Severity in Patients with NAFLD—A Prospective Case–Control Study"

_microorganisms, 2023, doi:10.3390/microorganisms11061571_

Round 1
Reviewer 1 Report
This article critically evaluates a case-control study that investigates the potential link between Non-alcoholic Fatty Liver Disease (NAFLD) and COVID-19 severity, with a specific focus on the role of Transforming Growth Factor-Beta 1 (TGF-β1) as a potential immunomodulatory biomarker. The study provides valuable insights into the association between NAFLD and COVID-19, shedding light on the immunological mechanisms that underlie the exacerbation of COVID-19 in patients with NAFLD. The findings suggest that TGF-β1 levels may serve as an efficient predictive biomarker for assessing disease severity and adverse outcomes in COVID-19 patients with NAFLD. However, several methodological and interpretational considerations need to be addressed to strengthen the study's conclusions.
1. Is the sample size of 60 patients, including 30 with NAFLD, sufficient to draw robust conclusions about the relationship between NAFLD, TGF-β1, and COVID-19 severity?
2. It is stated that TGF-β has a paradoxical dual role, promoting both immunosuppressive and pro-inflammatory effects. How does this dual role influence the immunological equilibrium, and how is it relevant to the pathophysiology of NAFLD and COVID-19?
3. The association between increased TGF-β levels and NAFLD severity is mentioned, but what are the potential mechanisms through which TGF-β contributes to the development and progression of liver disease?
4. Given that this study is part of a larger prospective cohort study, could you provide some information on the design and objectives of the COVID-FAT trial (ClinicalTrials.gov Identifier: NCT04982328)?
5. It is mentioned that patients were classified into three subgroups - mild, moderate, and severe COVID-19 - according to the NIH classification. Could you provide more information about the criteria used to classify patients into these categories?
6. Why was the term NAFLD and not MAFLD used for the work?
7. There is no clear definition of NAFLD in the introduction. Make it more clear.
8. In the sentence "In the pre-COVID era, the role of NAFLD in infectious diseases was largely overlooked [4,5], however, the pandemic has shed new light on the intertwined relationship between NAFLD and infection," consider expanding on how the pandemic has brought attention to the relationship between NAFLD and infection. Provide specific examples or studies that have contributed to this understanding.
9. Line 30, Nonalcoholic fatty liver disease" should be "Non-alcoholic fatty liver disease."
10. Line 369, in the sentence "We included a relatively small number of patients," it would be more precise to mention the actual number of patients included.
11. Elevated levels of TGF-β1 in chronic liver disease may be caused by the presence of fibrosis, as a consequence of persistent inflammation in NAFLD, and not by the presence of NAFLD per se.
12. Additional non-invasive tests could be performed to determine the extent of liver fibrosis in the NAFLD cohort.
13. Moderate English grammar and editing are needed.
Moderate English grammar and editing are needed
Author Response
REVIEWER 1
This article critically evaluates a case-control study that investigates the potential link between Non-alcoholic Fatty Liver Disease (NAFLD) and COVID-19 severity, with a specific focus on the role of Transforming Growth Factor-Beta 1 (TGF-β1) as a potential immunomodulatory biomarker. The study provides valuable insights into the association between NAFLD and COVID-19, shedding light on the immunological mechanisms that underlie the exacerbation of COVID-19 in patients with NAFLD. The findings suggest that TGF-β1 levels may serve as an efficient predictive biomarker for assessing disease severity and adverse outcomes in COVID-19 patients with NAFLD. However, several methodological and interpretational considerations need to be addressed to strengthen the study's conclusions.
- Is the sample size of 60 patients, including 30 with NAFLD, sufficient to draw robust conclusions about the relationship between NAFLD, TGF-β1, and COVID-19 severity?
Authors’ response: We thank the reviewer for the comment. We agree with the reviewer that this is an important study limitation, and this is clarified in limitations sections.
The number of 60 patients was selected based on a priori sample size calculation (now added in Methods).
- It is stated that TGF-β has a paradoxical dual role, promoting both immunosuppressive and pro-inflammatory effects. How does this dual role influence the immunological equilibrium, and how is it relevant to the pathophysiology of NAFLD and COVID-19?
Authors’ response: We have added the additional explanation in the Discussion section.
- The association between increased TGF-β levels and NAFLD severity is mentioned, but what are the potential mechanisms through which TGF-β contributes to the development and progression of liver disease?
Authors’ response: We thank the reviewer for this comment. This is now added in the Introduction section.
- Given that this study is part of a larger prospective cohort study, could you provide some information on the design and objectives of the COVID-FAT trial (ClinicalTrials.gov Identifier: NCT04982328)?
Authors’ response: We have added a brief explanation of the study aims.
- It is mentioned that patients were classified into three subgroups - mild, moderate, and severe COVID-19 - according to the NIH classification. Could you provide more information about the criteria used to classify patients into these categories?
Authors’ response: The patients were classified based on well-established NIH criteria on COVID-19 severity (https://www.covid19treatmentguidelines.nih.gov/overview/clinical-spectrum/). The was now briefly explained in the manuscript:
“Depending on the severity of the disease, according to the NIH classification, patients were classified into 3 subgroups – moderate (bilateral pneumonia with SpO2 > 93% on room air, severe (dyspnea and/or tachypnea > 24/min and/or SpO2 < 93% and critical COVID-19 (requiring intensive care unit care, criteria for ARDS, advanced respiratory support with HFNC, non- invasive/invasive mechanical ventilation).“
- Why was the term NAFLD and not MAFLD used for the work?
Authors’ response: We are aware of the ongoing debate and the new proposed definition of MAFLD (J Hepatol. 2020 Jul;73(1):202-209.), which requires presence of obesity and/or DM, and in patients with normal BMI at least two components of metabolic syndrome. However, this has not yet been included in official guidelines. Presently the definition of NAFLD as reported in most guidelines and recent publications is based on the presence of steatosis in >5% of hepatocytes in the absence of significant ongoing or recent alcohol consumption and other known causes of liver disease, and this is the definition of NAFLD that we used in our study.
- There is no clear definition of NAFLD in the introduction. Make it more clear.
Authors’ response: NAFLD definition and explanation of its importance is added in Introduction.
- In the sentence "In the pre-COVID era, the role of NAFLD in infectious diseases was largely overlooked [4,5], however, the pandemic has shed new light on the intertwined relationship between NAFLD and infection," consider expanding on how the pandemic has brought attention to the relationship between NAFLD and infection. Provide specific examples or studies that have contributed to this understanding.
Authors’ response: We thank the reviewer for this comment. We have added brief explanation in the Introduction.
- Line 30, Nonalcoholic fatty liver disease" should be "Non-alcoholic fatty liver disease."
Authors’ response: Corrected.
- Line 369, in the sentence "We included a relatively small number of patients," it would be more precise to mention the actual number of patients included.
Authors’ response: Corrected.
- Elevated levels of TGF-β1 in chronic liver disease may be caused by the presence of fibrosis, as a consequence of persistent inflammation in NAFLD, and not by the presence of NAFLD per se. Additional non-invasive tests could be performed to determine the extent of liver fibrosis in the NAFLD cohort.
Authors’ response: We thank the reviewer for this comment. Unfortunately, we could not assessed NAFLD severity in this cohort. We have added APRI and FIB-4 score in Table 1.
This is now added in study limitations:
“… Since fibrosis scores may not be reliable in hospitalized patients with COVID-19 which is frequently accompanied with elevated aminotransferases and elastography was not available at the COVID-19 department, we could not evaluate fibrosis stage, consequently, the effect of advanced NAFLD on TGF-β1 levels could not be analyzed…”
- Moderate English grammar and editing are needed.
Authors’ response: Grammar and spelling were checked.
Reviewer 2 Report
This is an interesting case-control study aiming to evaluate whether NAFLD patients have distinct TGF-β serum concentrations that might be used as a prognostic biomarker for COVID-19 severity. The authors included a total of 60 participants, 30 with NAFLD. Although the study is of interest and is comprehensively conducted, the subgroup analyses included a very limited number of participants limiting its statistical significance. I believe that several comments need to be addressed by the authors.
Methods:
· Describe any efforts to address potential sources of bias.
· What sampling method was used? Please mention how did you arrive to the study sample size.
· How was hepatic steatosis evaluated? Who performed the assessment? Was the investigator blinded to the study aims and patient data?
· Define the definition of excessive alcohol consumption.
· Describe what investigations were performed to diagnose and exclude other secondary causes of hepatic steatosis.
· Page 2, line 96: absolute neutrophil count (AND). The abbreviation should be corrected.
· Who performed the ultrasound measurements or measuring the visceral and subcutaneous abdominal fat thickness? Was the investigator blinded to the study aims and patient data?
· Regression analysis should be conducted. Key potential confounding variables that were measured should be adjusted statistically for their impact on the relationship between exposure(s) and outcome(s).
Results:
· Please report the numbers of individuals at each stage of study (e.g., number of participants potentially eligible, examined for eligibility, confirmed eligible, included in the study, completing follow-up, and analyzed). Also, give reasons for non-participation at each stage. You can consider the use of a flow diagram.
· I suggest adding a subgroup analysis evaluating TGF-β levels according to NAFLD severity.
Discussion:
Several study limitations need to be mentioned such as:
· Causality between the evaluated parameters can't be confirmed or negated due to the observational study design.
· Results can't be generalized due to the limited study population and uniform sample population.
· Subgroup analyses included a very small number of participants in each group limiting the statistical significance of the obtained results.
· Longitudinal studies are required to evaluate TGF-β levels prospectively regarding the assessed parameters.
Minor editing of English language required.
Author Response
This is an interesting case-control study aiming to evaluate whether NAFLD patients have distinct TGF-β serum concentrations that might be used as a prognostic biomarker for COVID-19 severity. The authors included a total of 60 participants, 30 with NAFLD. Although the study is of interest and is comprehensively conducted, the subgroup analyses included a very limited number of participants limiting its statistical significance. I believe that several comments need to be addressed by the authors.
Methods:
- Describe any efforts to address potential sources of bias.
Authors’ response: This was part of the prospective study that had strict inclusion and exclusion criteria to avoid selection bias. Known confounders that might be associated with liver steatosis were controlled. However, our study should be viewed within its limitations as stated and expanded at the end of the Discussion according to reviewers’ suggestions.
- What sampling method was used? Please mention how did you arrive to the study sample size.
Authors’ response: We performed a power analysis which is now included in the manuscript. The main aim of this study was to evaluate the difference in serum TGF-β1 concentrations between two groups of patients. Therefore, 60 consequently hospitalized COVID-19 patients (30 with and 30 without NAFLD) were included. This is now clarified in Methods. For the same reason we do not believe that study flow chart is needed.
- How was hepatic steatosis evaluated? Who performed the assessment? Was the investigator blinded to the study aims and patient data?
Authors’ response: Upon admission, both visceral and abdominal fat thickness and liver steatosis were assessed by ultrasound by an experienced radiologist and defined as increased echogenicity and sound attenuation of liver parenchyma. The radiologist was blinded to the study aims and patient data. This is now clarified.
- Define the definition of excessive alcohol consumption.
Authors’ response: We used the AASLD and US definition for significant alcohol consumption. Citation is added in the revised manuscript.
- Describe what investigations were performed to diagnose and exclude other secondary causes of hepatic steatosis.
Authors’ response: By carefully selecting the exclusion criteria mentioned in the text, we tried to exclude all potential secondary causes of NAFLD. Patients were systematically screened for viral hepatitis B and C, and alcohol consumption was assessed based on anamnestic data. Patients on chronic therapy known to be associated with liver steatosis (e.g. amiodarone, antiepileptics) were excluded. This is now clarified in the text.
- Page 2, line 96: absolute neutrophil count (AND). The abbreviation should be corrected.
Authors’ response: Corrected.
- Who performed the ultrasound measurements or measuring the visceral and subcutaneous abdominal fat thickness? Was the investigator blinded to the study aims and patient data?
Authors’ response: Answered above.
- Regression analysis should be conducted. Key potential confounding variables that were measured should be adjusted statistically for their impact on the relationship between exposure(s) and outcome(s).
Authors’ response: The regression analysis is now included in the manuscript.
Results:
- Please report the numbers of individuals at each stage of study (e.g., number of participants potentially eligible, examined for eligibility, confirmed eligible, included in the study, completing follow-up, and analyzed). Also, give reasons for non-participation at each stage. You can consider the use of a flow diagram.
Authors’ response: As described above, we do not think that flow-chart is needed.
- I suggest adding a subgroup analysis evaluating TGF-β levels according to NAFLD severity.
Authors’ response: We thank the reviewer for this comment. Unfortunately, we could not assess NAFLD severity in this cohort.
This is now added in study limitations:
“… Since fibrosis scores may not be reliable in hospitalized patients with COVID-19 which is frequently accompanied with elevated aminotransferases and elastography was not available at the COVID-19 department, we could not evaluate fibrosis stage, consequently, the effect of advanced NAFLD on TGF-β1 levels could not be analyzed…”
Discussion:
Several study limitations need to be mentioned such as:
1.) Causality between the evaluated parameters can't be confirmed or negated due to the observational study design.
2.) Results can't be generalized due to the limited study population and uniform sample population.
3.) Subgroup analyses included a very small number of participants in each group limiting the statistical significance of the obtained results.
4.) Longitudinal studies are required to evaluate TGF-β levels prospectively regarding the assessed parameters.
Authors’ response: We agree with the reviewer. These limitations are now added in revised version of the manuscript.
Reviewer 3 Report
Overall an interesting manuscript that tries to shed some light in the role of TGF-b in COVID-19. However I have some questions/comments
1. In materials and methods section, line 73, you state you have excluded patients that were admitted to the ICU or died within the first 48 hours of hospitalization. Since these patients are the most severely ill an explanation for the fact that you excluded them must be given. Otherwise you should include them in the study
2. When reading patients' characteristics I see that you had 6 patients with DM and 10 obese with no NAFLD. Since the vast majority of DM and obese patients have NAFLD, it would be interesting to have a comment about the absence of NAFLD in these patients
3. Also in patients characteristics section. I understand you had no cirrhotics in your cohort. This should also be mentioned
4. In lines 159-162 you show that the presence of NAFLD didn't lead to worse COVID-19-related outcomes. This should be briefly discussed in discussion section
5. In results section, fig 3, I don't see a statistical correlation between TGF-b and IL-6. Since IL-6 seems to be a key factor for patietns with severe COVID-19 it would be interesting to see a statistical correlation between those 2
6. Also in results section, a multivariable analysis regarding COVID-19 progression is missing
7. In line 364, regarding the role of TGF-b in NAFLD progression more references should be added (t.ex. Ahmed H, et al. Exp Mol Pathol. 2022 Feb;124:104733. doi: 10.1016/j.yexmp.2021.104733; Kumar S, et al, Adv Drug Deliv Rev. 2021 Sep;176:113869. doi: 10.1016/j.addr.2021.113869 and Nasiri-Ansari N, et al, Cells. 2022 Aug 12;11(16):2511. doi: 10.3390/cells11162511)
Good use of english language
Author Response
Overall an interesting manuscript that tries to shed some light in the role of TGF-b in COVID-19. However I have some questions/comments
- In materials and methods section, line 73, you state you have excluded patients that were admitted to the ICU or died within the first 48 hours of hospitalization. Since these patients are the most severely ill an explanation for the fact that you excluded them must be given. Otherwise you should include them in the study.
Authors’ response: We thank the reviewer for this comment. The patients who were admitted to the ICU and/or died within the first 48 hours of hospitalization were not included for several reasons. This was part of the CovidFAT study that was designed to analyze the impact of NAFLD on COVID-19 outcomes. Since there was possibility that patients with early mortality could not be timely evaluated and screened for MetS and NAFLD, this could lead to significant selection bias. Therefore, this group of patients was excluded. Next practical reason was that due to the work organization during the COVID-19 pandemic, it was not possible to timely include ICU patients at the time of this study.
This is now added in study limitations. - When reading patients' characteristics I see that you had 6 patients with DM and 10 obese with no NAFLD. Since the vast majority of DM and obese patients have NAFLD, it would be interesting to have a comment about the absence of NAFLD in these patients.
Authors’ response: We agree with the reviewer that NAFLD is strongly associated with other components of metabolic syndrome, including obesity. The incidence of NAFLD among obese subjects varies around 50-90%. In non-obese persons, NAFLD is frequently associated with other components of metabolic syndrome, and in up to 25% there is no clear association. Due to the limited number of included patients we were not able to perform subanalysis based on the presence of MetS components.
- Also in patients characteristics section. I understand you had no cirrhotics in your cohort. This should also be mentioned.
Authors’ response: Patients with liver cirrhosis were excluded. Unfortunately, we could not assess NAFLD severity in this cohort. This is now added in study limitations. - In lines 159-162 you show that the presence of NAFLD didn't lead to worse COVID-19-related outcomes. This should be briefly discussed in discussion section
Authors’ response: This study was not designed and powered to detect difference in mortality, but TGF-β1 difference between these 2 groups. As suggested, we briefly discussed it in Discussion.
- In results section, fig 3, I don't see a statistical correlation between TGF-b and IL-6. Since IL-6 seems to be a key factor for patietns with severe COVID-19 it would be interesting to see a statistical correlation between those 2
Authors’ response: We agree with the authors. However, there was no significant correlation between serum concentrations of IL-6 and TGF-β1 as shown in Figure 3.
- Also in results section, a multivariable analysis regarding COVID-19 progression is missing
Authors’ response: We added multivariable analysis.
- In line 364, regarding the role of TGF-b in NAFLD progression more references should be added (t.ex. Ahmed H, et al. Exp Mol Pathol. 2022 Feb;124:104733. doi: 10.1016/j.yexmp.2021.104733; Kumar S, et al, Adv Drug Deliv Rev. 2021 Sep;176:113869. doi: 10.1016/j.addr.2021.113869 and Nasiri-Ansari N, et al, Cells. 2022 Aug 12;11(16):2511. doi: 10.3390/cells11162511)
Authors’ response: We thank reviewer for suggesting these references. They are included in the manuscript.
Round 2
Reviewer 1 Report
Accept in present form
Reviewer 2 Report
The authors modified the manuscript according to my previously mentioned comments. I believe that the manuscript has been properly modified and can be considered for publication.
Minor editing of English language required